# Evaluation of Fengyun-3C Soil Moisture Products Using In-Situ Data from the Chinese Automatic Soil Moisture Observation Stations: A Case Study in Henan Province, China

**Yongchao Zhu [1,2]** , **Xuan Li [1,2]**, **Simon Pearson [3]** , **Dongli Wu [4]**, **Ruijing Sun [5]**, **Sarah Johnson [6]**, **James Wheeler [6]** and **Shibo Fang [1,2,*]**

[1]  State Key Laboratory of Severe Weather, Chinese Academy of Meteorological Sciences, Beijing 100081, China; ychzhuvip@yeah.net (Y.Z.); xuanli@ceh.ac.uk (X.L.)

[2]  Collaborative Innovation Centre on Forecast and Evaluation of Meteorological Disasters, Nanjing University of Information Science & Technology, Nanjing 210044, China

[3]  Lincoln Institute for Agri-Food Technology, University of Lincoln, Lincoln LN67TS, UK; SPearson@lincoln.ac.uk

[4]  Chinese Meteorological Observation Centre, Meteorological Administration Meteorological, Beijing 100081, China; wudongli@cma.gov.cn

[5]  National Satellite Meteorological Centre, China Meteorological Administration, Beijing 100081, China; sunrj@cma.gov.cn

[6]  Centre for Landscape and Climate Research, School of Geography, Geology and the Environment, University of Leicester, Leicester LE17RH, UK; sj239@leicester.ac.uk (S.J.); jemw3@leicester.ac.uk (J.W.)

*  Correspondence: fangshibo@cma.gov.cn; Tel.: +86-15801635580

**Abstract:** Soil moisture (SM) products derived from passive satellite missions are playing an increasingly important role in agricultural applications, especially crop monitoring and disaster warning. Evaluating the dependability of satellite-derived soil moisture products on a large scale is crucial. In this study, we assessed the level 2 (L2) SM product from the Chinese Fengyun-3C (FY-3C) radiometer against in-situ measurements collected from the Chinese Automatic Soil Moisture Observation Stations (CASMOS) during a one-year period from 1 January 2016 to 31 December 2016 across Henan in China. In contrast, we also investigated the skill of the Advanced Microwave Scanning Radiometer 2 (AMSR2) and Soil Moisture Active/Passive (SMAP) SM products simultaneously. Four statistical parameters were used to evaluate these products' reliability: mean difference, root-mean-square error (RMSE), unbiased RMSE (ubRMSE), and the correlation coefficient. Our assessment results revealed that the FY-3C L2 SM product generally showed a poor correlation with the in-situ SM data from CASMOS on both temporal and spatial scales. The AMSR2 L3 SM product of JAXA (Japan Aerospace Exploration Agency) algorithm had a similar level of skill as FY-3C in the study area. The SMAP L3 SM product outperformed the FY-3C temporally but showed lower performance in capturing the SM spatial variation. A time-series analysis indicated that the correlations and estimated error varied systematically through the growing periods of the key crops in our study area. FY-3C L2 SM data tended to overestimate soil moisture during May, August, and September when the crops reached maximum vegetation density and tended to underestimate the soil moisture content during the rest of the year. The comparison between the statistical parameters and the ground vegetation water content (VWC) further showed that the FY-3C SM product performed much better under a low VWC condition (<0.3 kg/m$^2$) than a high VWC condition (>0.3 kg/m$^2$), and the performance generally decreased with increased VWC. To improve the accuracy of the FY-3C SM product, an improved algorithm that can better characterize the variations of the ground VWC should be applied in the future.

**Keywords:** soil moisture; Fengyun-3C; passive microwave; Chinese Automatic Soil Moisture Observation Stations; NDVI

## 1. Introduction

Soil moisture (SM) is one of the fundamental environmental variables in the global energy and water cycles [1]. As satellite-based soil moisture products have become more widely available, they have played increasingly important roles in many applications, such as meteorology, hydrology, climatology, and agriculture [2]. Accurate measurement of soil moisture on large scales may assist in crop yield estimation, drought prediction, and disaster monitoring in agricultural regions, particularly in arid and semiarid areas where regular irrigation is required but water resources are limited.

Surface soil moisture can be obtained by various means, such as in situ soil moisture instruments, land surface models, and remote sensing technology [3]. Since the first passive microwave satellite sensor, launched in 1978, various studies have demonstrated that it is feasible to retrieve soil moisture from passive microwave satellite missions [4]. Passive microwave satellite missions have been widely used for soil moisture estimation, such as the Soil Moisture and Ocean Salinity (SMOS) mission [5,6], the Soil Moisture Active/Passive (SMAP) mission [7], the Special Sensor Microwave/Imager (SSM/I) mission, the Advanced Microwave Scanning Radiometer for the Earth Observing System (AMSR-E) [8,9], the Advanced Microwave Scanning Radiometer 2 (AMSR2) mission [10], and a series of China's Fengyun 3 (FY-3) satellites, consisting of FY-3A, FY-3B, FY-3C, and FY-3D [11–13].

Since soil moisture products are generally based on different satellite data and algorithms, their quality and continuity vary in space and time [14]. Validation is an important task for any satellite-based soil moisture product, as it not only aids appraisal of the actual accuracy of the delivered soil moisture estimates but also improves our understanding of the product's advantages and disadvantages under different ground conditions and temporally [15]. Numerous studies have assessed the accuracy of the soil moisture products from SMOS, SMAP, AMSR-E, and AMSR2 by comparing the estimations against the ground measurements from monitoring networks around the world [2,3,5,16–21].

The FY-3 satellite series is China's second-generation polar-orbiting satellite series and includes four satellites, FY-3A, FY-3B, FY-3C, and FY-3D, with an approximate two-year separation between two subsequent launches [11]. The first two experimental satellites, FY-3A and FY-3B, were successfully launched on 27 May 2008 and 5 December 2010, respectively, whereas FY-3C and FY-3D were respectively sent into orbit on 23 September 2013 and 14 November 2017 [22]. FY-3A and FY-3C orbit midmorning with their local solar time on descending node (LTDN) around 10:00 a.m., whereas FY-3B and FY-3D orbit in the afternoon with their local solar time on ascending node (LTAN) around 10:00 p.m. [23]. A similar microwave radiation imager (MWRI) was aboard FY-3B, FY-3C, and FY-3D, which observes the Earth's surface at five different microwave frequencies ranging from 10 to 89 GHz. The MWRI can complete the coverage of the Earth's surface within two to three days, with a swath of 1400 km. Observations from the MWRI have been used to retrieve land surface parameters, such as soil moisture, vegetation water content, and land surface temperature.

An official soil moisture product derived from the MWRI observations was distributed by the National Satellite Meteorological Centre (NSMC) of China, which is available for all registered users (http://satellite.nsmc.org.cn/portalsite/default.aspx). To obtain the official soil moisture product from the MWRI-observed brightness temperatures, NSMC used a modified Single Channel Algorithm (SCA) proposed by Jackson [24]. The significant differences between the retrieval algorithm and other algorithms are that the FY-3 algorithm utilizes a new surface emission model (the $Q_p$ Model) [25,26] to correct the effects of surface roughness and the algorithm uses both vertical and horizontal polarizations of the X-band (10.65 GHz) brightness temperatures to retrieve the soil moisture instead of one single channel. Working on the X-band, the FY-3 SM product is expected to sense and record soil moisture

content contained in the top ~1 cm of the soil layer, on average, for low-vegetated areas [3,27]. However, to the best of our knowledge, limited research has focused on evaluating the accuracy of the soil moisture product from FY-3 series satellites. Parinussa et al. [13] first compared the soil moisture products derived from the FY-3B official algorithm and the land parameters retrieval model (LPRM) against in-situ measurements. Their results indicated that the two products could both capture the temporal variation of soil moisture well at nighttime. The best agreement with in-situ measurements was found in sparsely to moderately vegetated regions, and the agreement was less reliable with increased vegetation density. Cui, et al. [28] conducted a detailed examination of the quality of the FY-3B soil moisture products along with other seven soil moisture products from different satellites. They found that the FY-3B soil moisture product exhibited a good temporal performance against in-situ measurements collected from two soil moisture network regions in the United States and Spain.

Since the soil moisture product is not available currently for FY-3D, the FY-3C soil moisture product is generally believed to have the best observation accuracy. In this paper, we evaluated the level 2 (L2) soil moisture product from the FY-3C MWRI over Henan province, a key agricultural region where the crop rotation consists mainly of winter wheat and summer maize. In contrast, we also investigated the skill of the AMSR2 (Japan Aerospace Exploration Agency (JAXA) algorithm) level 3 (L3) soil moisture product and the SMAP L3 passive soil moisture product over this agricultural region. To assess the performance of the three soil moisture products above, we compared the products with in situ soil moisture data from 113 monitoring stations from the Chinese Automatic Soil Moisture Observation Stations (CASMOS) network over a one-year period from 1 January 2016 to 31 December 2016 at daily and monthly time scales. We evaluated the soil moisture product using four statistical parameters: the mean difference (MD), the root-mean-square error (RMSE), the unbiased RMSE (ubRMSE), and the correlation coefficient (R). Additionally, we analyzed how the cropping system affected the FY-3C soil moisture product's performance over the region. At last, possible error sources in the FY-3C soil moisture product are also investigated and discussed. This paper is structured as follows. Section 2 introduces the study area, the satellite soil moisture products, the in-situ measurements, and other ancillary datasets. Section 3 describes the evaluation method used in this study. Section 4 presents the comparison results between the satellite products and their corresponding in-situ measurements. In Section 5, the possible error sources in the soil moisture products will be discussed. Section 6 draws the conclusions of this paper.

## 2. Study Area and Datasets

### 2.1. Study Area

Henan province, located in the middle part of China (Figure 1b), is one of the most important granaries in the country, extending from 31°23' N to 36°22' N and 110°21' E to 116°39' E, with an area of $16.7 \times 10^4$ km$^2$ and an average elevation of 100 m above sea level. Henan has a typical temperate monsoon climate and an annual mean temperature of 10–15 °C. The yearly precipitation is unevenly distributed among seasons, roughly ranging from 400 to 800 mm, and more than 50% of precipitation events occur in the summer during the maize growing season. Both temperature and precipitation decrease gradually from southeast to northwest. As shown in Figure 1a, nearly half of the region is planted with crops, and the prevailing double-cropping system is winter wheat and summer maize. Generally, the wheat growing season is from the early October to the next June. The corn growing season is from June to late September. Due to insufficient precipitation in spring and winter, supplemental irrigation for winter wheat is required to obtain optimum yields. As a result, Henan suffers from severe water shortages and environmental problems related to groundwater overexploitation. Therefore, strengthening soil moisture monitoring in the agriculture areas is of great significance for improving water use efficiency in this province.

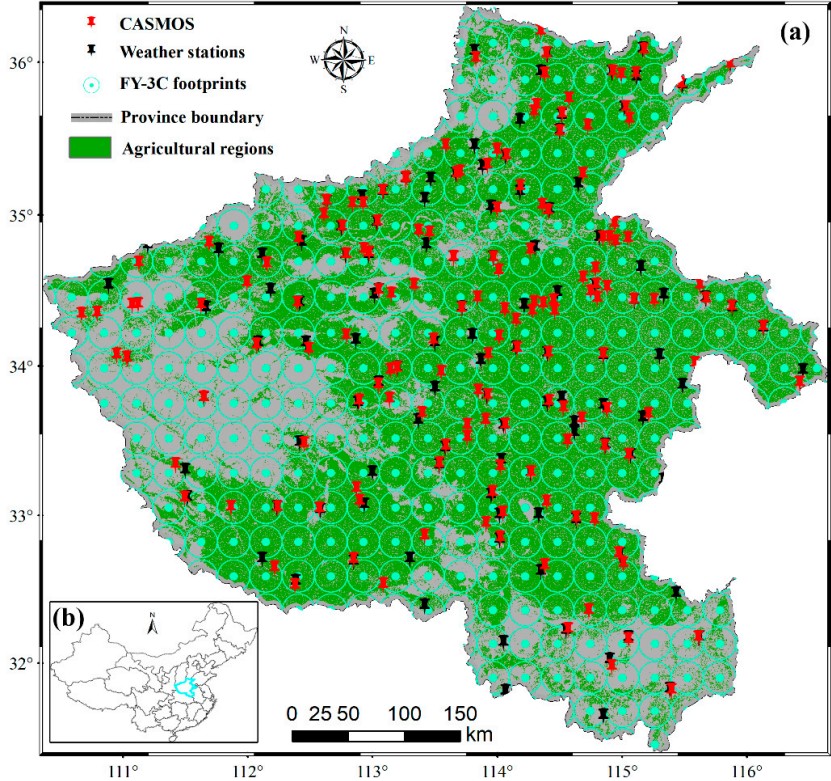

**Figure 1.** (**a**) The agricultural regions, Chinese Automatic Soil Moisture Observation Stations (CASMOS), the weather stations, and the footprints of Fengyun-3C (FY-3C) located in Henan province; (**b**) The location of Henan province in China.

Relative to soil moisture, soil type per se is considered to have a small yet significant impact on the brightness temperatures observed at the satellite footprint scale [9]. Consequently, soil texture is usually approximated as constant in soil moisture retrieval algorithms. Figure 2 shows the soil types in Henan at a 1:1,000,000 scale sourced from the Second National Soil Survey of China [29]. It was classified using the Genetic Soil Classification of China (GSCC) [30]. The predominant soil types are Alfisols, Semi-Alfisols, and Semi-Hydromorphic soils, which occupy approximately 83% of the area of this region. Table 1 gives their areal fraction and soil separate compositions of the three main soil types.

**Table 1.** The main soil types and their soil separates in Henan. Soil particles are grouped according to their size into what are called soil separates (clay, silt, and sand). The soil diameter limits for clay, silt, and sand are less than 0.002 mm, 0.002–0.02 mm, and 0.02–2 mm, respectively.

| Soil Type (Order) | Areal Fraction | Clay (%) | Silt (%) | Sand (%) |
|---|---|---|---|---|
| Alfisols | 23.4% | 10.8–19.4 | 18.5–32.6 | 34.0–60.2 |
| Semi-Alfisols | 18.0% | 16.7–25.3 | 25.4–36.6 | 37.2–65.6 |
| Semi-Hydromorphic soils | 41.6% | 25–35.4 | 20.0–37.6 | 35.4–54.0 |
| Others | 17.0% | - | - | - |

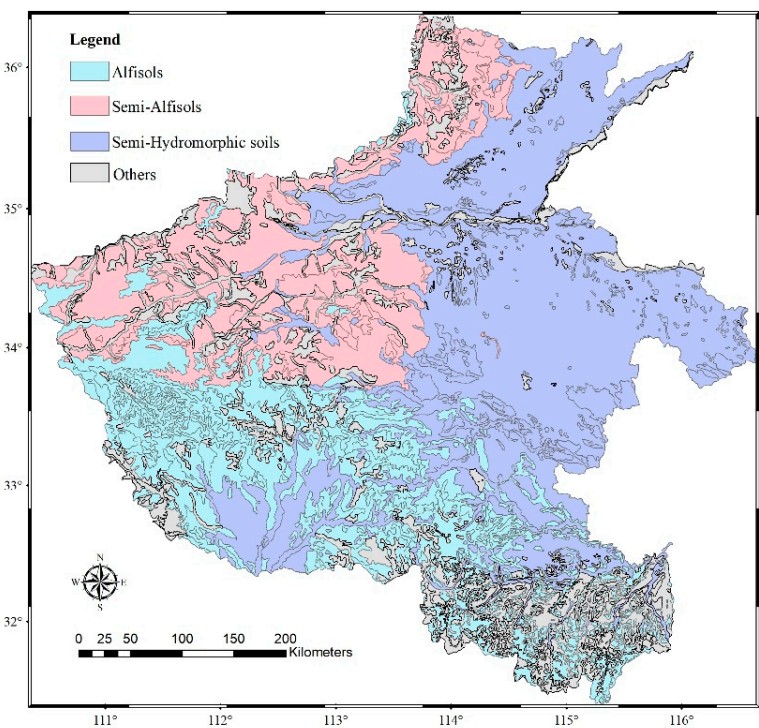

**Figure 2.** The soil types map in Henan. There are eight orders, 23 great groups, and 59 sub-great groups. Herein, only the three main orders in Henan are explicitly displayed.

*2.2. FY-3C L2 Soil Moisture Product*

The MWRI aboard the FY-3C observes the Earth's surface on 10 channels ranging from 10.7 GHz to 89.0 GHz. The band information for the MWRI aboard FY-3C is listed in Table 2. The FY-3 L2 soil moisture products are described in volumetric water content in $m^3/m^3$, which are retrieved from the brightness temperatures collected by the MWRI using the radiative transfer model [13]. The FY-3C L2 soil moisture products are available from May 2014 to present (2019). They include three products with different time scales: daily, 10-day average, and monthly average, each of which separately consists of two subsets: one from the ascending orbits (10:00 p.m. local time) and the other from descending orbits (10:00 a.m. local time). In this study, the daily product, combining both the ascending and descending datasets, was used. If the two datasets overlapped, their averaged value was used. All the L2 soil moisture products are posted on a 25-km Equal-Area Scalable Earth-1 (EASE1) grid [31], and the footprints of the products in our study region are plotted in Figure 1a. According to the documents of the products, FY-3C L2 SM products provide the amount of soil moisture of the top 5-cm layer, with the desired estimation accuracy of 0.06 $m^3/m^3$ [13].

**Table 2.** Introduction to the microwave radiation imager channels.

| Frequency (GHz) | Polarization | Bandwidth (MHz) | Sensitivity (K) | IFOV km × km | Pixel Size km × km |
|---|---|---|---|---|---|
| 10.65 | V/H | 180 | 0.5 | 51 × 85 | 40 × 11.2 |
| 18.70 | V/H | 200 | 0.5 | 30 × 50 | 40 × 11.2 |
| 23.80 | V/H | 400 | 0.8 | 27 × 45 | 20 × 11.2 |
| 36.50 | V/H | 900 | 0.5 | 18 × 30 | 20 × 11.2 |
| 89.00 | V/H | 4600 | 1.0 | 9 × 15 | 10 × 11.2 |

Note: V, vertical polarization; H, horizontal polarization; IFOV, instantaneous field of view.

The current FY-3C SM retrieval algorithm is a radiative transfer-based model that links soil moisture, land surface temperature, and vegetation optical depth to brightness temperature observations ($T_b$) observed by the MWRI [32]. The parameters in the algorithm are summarized in Table 3. First, the algorithm assumes that soil temperature ($T_s$) and vegetation canopy temperature

$(T_c)$ are equal and estimates the surface temperature based on a linear relationship between vertical polarization $T_b$ at 36.5 GHz. Then, the algorithm connects the emissivity with the surface roughness using a parameterized bare surface emission model (the $Q_p$ model), which takes into account the effects of the surface roughness on the emission signals through the roughness variable $Q_p$ at different polarizations $p$ [25]. The $Q_p$ can be simply described as a function of the ratio of the surface root-mean-square height and the correlation length. Next, the algorithm uses the empirical relationship between the Normalized Difference Vegetation Index (NDVI), vegetation water content (VWC, $W_c$), and vegetation optical depth ($\tau$) to estimate $\tau$ [4]. The NDVI is a 10-day composite product calculated from the National Oceanic and Atmospheric Administration (NOAA) Advanced Very High-Resolution Radiometer (AVHRR). The algorithm uses the brightness temperature with both vertical and horizontal polarizations of 10.65 GHz to eliminate the effects of surface roughness and vegetation simultaneously. At last, the dielectric mixing model proposed by Wang and Schmugge [33] is used in the algorithm to convert the mixed dielectric constant to a soil moisture value. The ancillary data used during the retrieval process include global land surface classification data and soil texture data.

**Table 3.** Summary of the FY-3C soil moisture retrieval algorithm.

| Parameters | FY3C MWRI SM Retrieval Algorithm |
|---|---|
| Soil and vegetation canopy physical temperatures | $T_s = T_c$, linearly related with $T_b$ (36.5 GHz) |
| Surface roughness | $\log\left[Q_p(f)\right] = a_p(f) + b_p(f)\cdot\log(s/l) + c_p(f)\cdot(s/l)$ |
| Vegetation | $\tau = b\cdot W_c/\cos\theta$<br>$W_c = 5.0\cdot NDVI^2\ (NDVI > 0.5)$<br>$W_c = 2.5\cdot NDVI\ (NDVI \leq 0.5)$<br>$b = 0.28–0.33$, depending on the land type<br>$\omega = 0$ |
| Dielectric mixing model | Wang and Schmugge [33] |

Note: $T_s$, soil surface temperature; $T_c$, vegetation canopy temperature; $Q_p$, roughness parameters; the parameters $a_p(f)$, $b_p(f)$, and $c_p(f)$ depending on the frequency f and polarization p for the given microwave radiation imager (MWRI) incidence angle; s, root-mean-square height; l, correlation length; $\tau$, vegetation optical depth; $W_c$, vegetation water content; b, vegetation parameter; and $\omega$, single scattering albedo. NDVI, Normalized Difference Vegetation Index; SM, soil moisture.

### 2.3. AMSR2 and SMAP Soil Moisture Products

The AMSR2 level 3 (L3) daily soil moisture products collected during ascending and descending overpasses at 25-km resolution were used for evaluation. The data are available from August 2012 to present (2019). AMSR2 onboard the Global Change Observation Mission 1-Water (GCOM-W1) satellite was launched by the Japan Aerospace Exploration Agency (JAXA) in May 2012 [10]. The available soil moisture products derived from both the ascending (1:30 p.m. local time) and descending (01:30 a.m. local time) overpasses were provided by the JAXA Earth Observation Research Center (EORC). The soil moisture products are produced on daily and monthly time scales, and the spatial resolution is 0.1 degree (10 km) and 0.25 degree (25 km). These data are available for any registered user from JAXA (https://gcom-w1.jaxa.jp/). A radiative transfer-based model was used to produce the AMSR2 soil moisture product. Full details about the retrieval algorithm can be found in Fujii, et al. [34]. The soil moisture product from the nighttime (descending) overpass is generally expected to be more accurate than that from the daytime (ascending) overpass [2,3,35].

To match the spatial resolution of FY-3C and AMSR2 SM products, the daily SMAP passive level 3 product (version 5) with a spatial resolution of 36 km, generated on EASE-Grid 2.0, was chosen for evaluation in this study. The SMAP satellite was launched by the National Aeronautics and Space Administration (NASA) in January 2015 [7]. An L-band radar and an L-band radiometer were carried aboard the satellite. The local equatorial overpass time of the SMAP satellite is 6:00 p.m. and 6:00 a.m. for ascending and descending, respectively. SMAP measurements can provide direct sensing of soil moisture in the top 5 cm of the soil column with an accuracy of 0.04 $m^3/m^3$, which covers the globe every 2–3 days [36]. SMAP provides four different kinds of remotely sensed soil moisture products:

the passive, the active, the active-passive, and the enhanced passive soil moisture product, in which the SMAP passive soil moisture product that is available from 31 March 2015 to the present was used for evaluation. These SMAP products are freely available from the National Snow and Ice Data Center (NSIDC) (https://nsidc.org/data/smap/smap-data.html). The V-pol single channel algorithm (SCA-V) is the current baseline retrieval algorithm of the SMAP passive soil moisture product [28,37]. Refer to O'Neill, et al. [38] for more details about the SCA-V algorithm.

*2.4. In Situ Soil Moisture Measurements*

To improve the ability of drought monitoring and early disaster warning for the agricultural regions in China, since 2009, an extensive national soil moisture collecting network, CASMOS, has been developed by the Chinese Meteorological Administration (CMA) [39]. After several years of construction, more than 2000 observation stations have been set up in the agricultural areas of the country. Most of the observations contain eight measurement depths: 0–10, 10–20, 20–30, 30–40, 40–50, 50–60, 70–80, and 90–100 cm. The elements observed include soil volumetric water content, relative soil humidity, soil weight water content, and soil available water storage. Three types of observation instruments, DNZ1, DNZ2, and DNZ3, are separately produced by Shanghai Changwang Meteorological Science and Technology Corporation (Shanghai, China), Henan Meteorological Science Research Institute and the 27th Institute of China National Electric Power Corporation (Zhengzhou, China), and China Huayun Technology Development Corporation (Beijing, China) [40]. The operating principle of these three types of instruments is based on the frequency reflection method. DNZ1, which was set up in Henan, uses the standing wave method, whereas DNZ2 and DNZ3 use the capacitance method [41].

There are 158 monitoring stations in Henan province in total, in which the soil moisture measurements of 113 monitoring stations were picked out to validate the satellite soil moisture products in this paper. As displayed in Figure 1a, the observation stations cover more than 120 counties in the region and form an effective soil moisture monitoring network for the agricultural area [42]. The Meteorological Observation Centre of the CMA is responsible for data archiving and distribution. The CMA records the measurements every hour, and then the daily averaged values are produced from these data. For comparison with the depth of FY-3C soil moisture, we only used the soil moisture data from the 0–10-cm layer.

*2.5. MODIS NDVI and Precipitation Data*

The Moderate Resolution Imaging Spectroradiometer (MODIS) Normalized Difference Vegetation Index (NDVI), produced on 16-day intervals and at several spatial resolutions, enables consistent spatial and temporal comparisons of vegetation canopy greenness, which is a composite property of leaf area, chlorophyll, and canopy structure [43]. In this study, we used the MODIS NDVI product consisting of MYD13Q1 from the Aqua satellite and MOD13Q1 from the Terra satellite, both of which were retrieved from daily, atmosphere-corrected, and bidirectional surface reflectance with a spatial resolution of 250 m [44,45]. As the MODIS sensors aboard these two satellites are identical, the NDVI algorithm generates each 16-day composite 8 days apart, which permits a higher temporal resolution product by combining both products.

Figure 3 displays how the averaged NDVI and VWC of all the 113 CASMOS stations varied during a one-year period. We used a robust relationship (Equations (1) and (2)) between NDVI and VWC, proposed by Gao, et al. [46], to estimate the VWC in our study area. As shown in the figure, higher NDVI and VWC values corresponded with the crop growing periods with larger biomass; for winter wheat, these months were April and May, whereas, for summer maize, these months were July, August, and September.

$$VWC = 0.078 \cdot e^{3.510 \cdot NDVI}, \text{ for wheat} \tag{1}$$

$$VWC = 0.098 \cdot e^{4.225 \cdot NDVI}, \text{ for corn} \tag{2}$$

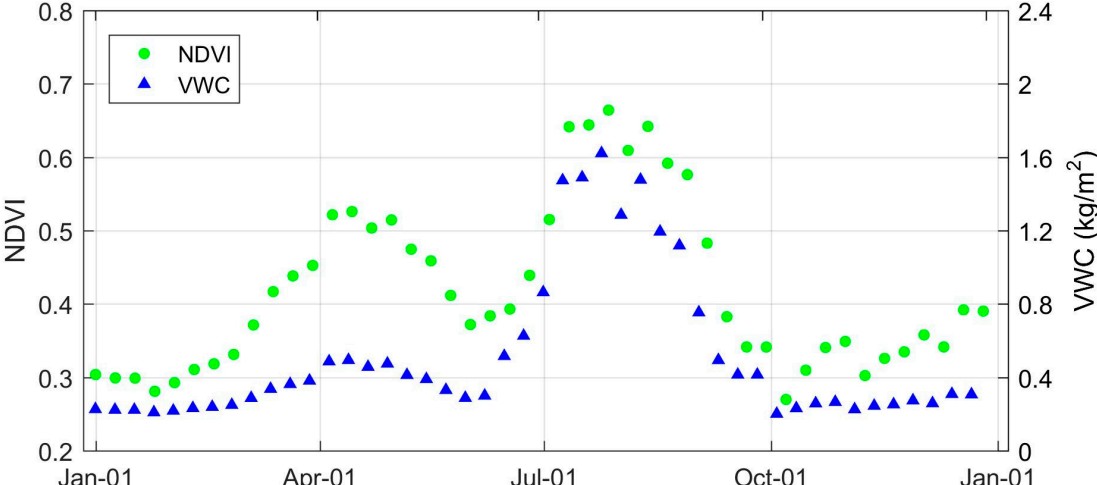

**Figure 3.** The averaged Normalized Difference Vegetation Index (NDVI) of all the CASMOS stations in 2016. The green circles represent the averaged NDVI from the Aqua and Terra satellites; the blue triangles represent the averaged vegetation water content (VWC) calculated from the NDVI data. Again, the growing seasons of wheat and corn are from early October to the next June and from June to late September, respectively.

Precipitation events are the most critical factors determining the surface soil moisture, and precipitation data can assist in validating the soil products derived from satellites [47]. We extracted the precipitation data from the China National Surface Weather Station Normalized Precipitation Dataset Version 3.0, which is archived by the Chinese Meteorological Data Service Centre (CMDSC, http://data.cma.cn). As shown in Figure 1a, in Henan province, there are 119 national weather collecting stations. In the dataset, the rainfall data are provided in mm/hour and daily averaged rainfall was further obtained based on the original data.

## 3. Methodology

As introduced above, many previous studies have evaluated different satellite-based soil moisture products using in situ soil moisture measurements [2,3,5,17,21]. In this paper, with the assistance of in situ soil moisture data from CASMOS and other auxiliary datasets, including rainfall and NDVI, we assessed the performance of the FY-3C L2 SM product in the agricultural regions of Henan province and analyzed factors that influence the results. At the same time, to compare with the skill of the FY-3C SM product, we also evaluated the performance of the AMSR2 and SMAP SM products. We employed four statistics to verify the effectiveness of the FY-3C, AMSR2, and SMAP products: the mean difference (MD), the root-mean-square error (RMSE), the unbiased RMSE (ubRMSE), and the correlation coefficient (R).

### 3.1. Study Framework and Data Integration

Figure 4 summarizes the workflows of this analytical framework. As shown in the figure, aside from the soil moisture data from the FY-3C MWRI, the five datasets introduced in Section 2—soil moisture products from AMSR2 and SMAP, in situ soil moisture measurements from CASMOS, rainfall from weather stations, and NDVI from MODIS—were integrated into the framework as well. However, these six datasets were different in both their spatial and temporal scales and in their sensing depths as well. For example, the soil moisture data from CASMOS and rainfall data were point measurements and daily averaged data were available nearly every day during the evaluation period. For the satellite soil moisture products of FY-3C, AMSR2, and SMAP, their spatial resolutions were 25 km, 25 km, and 36 km, respectively, with a similar temporal interval of 2–3 days. The resolution of MODIS NDVI was

250 m, and the temporal range was 8 days. Thus, the manner in which these datasets were integrated was crucial for the soil moisture products evaluation and later analysis.

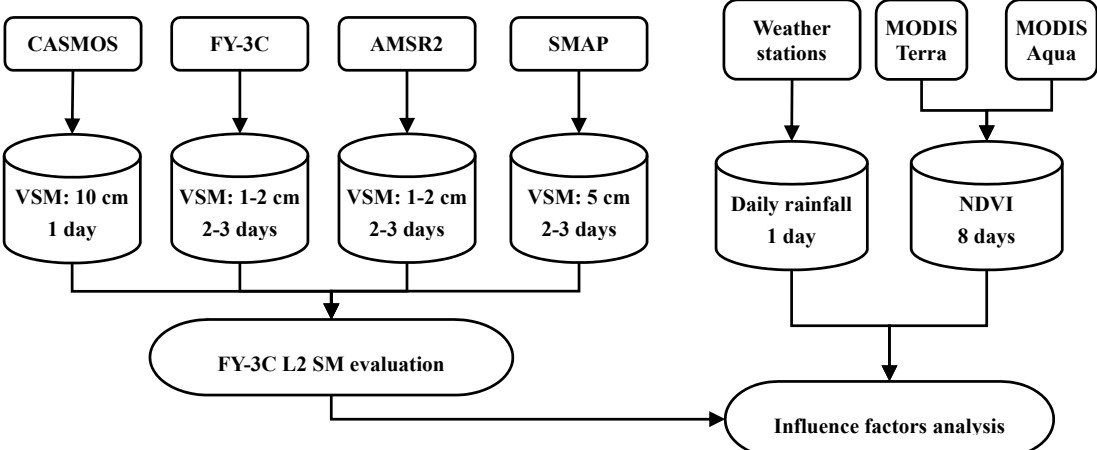

**Figure 4.** Work flowchart of this study. CASMOS represents the Chinese Automatic Soil Moisture Observation Stations deployed in the study area; Fengyun-3C (FY-3C), Advanced Microwave Scanning Radiometer 2 (AMSR2), and Soil Moisture Active/Passive (SMAP) represent the corresponding soil moisture products derived from the satellites; and VSM indicates the volumetric soil moisture with the unit in $m^3/m^3$.

When integrating the datasets, only those footprints that contained monitoring stations were used for evaluation. We generally extracted other datasets on the spatial scale based on the footprints of the FY-3C L2 soil moisture product (Figure 1a). In-situ data, including the soil moisture and the rainfall measurements, which lay within an FY-3C soil moisture footprint, were considered the corresponding ground truth for the region. If there was more than one station within a footprint, the averaged values were used. The AMSR2 and SMAP soil moisture products were resampled to the same grid size of FY-3C. When extracting the NDVI data to match the resolution of the FY-3C soil moisture product, all of the NDVI values within a footprint were averaged. Notably, some of the CASMOS were located in cities. These stations cannot correctly reflect the soil moisture information of the surrounding agricultural areas, and the satellite inversion results are profoundly affected by the buildings. Therefore, the 45 observation stations located in cities were excluded. Temporally, only the dates when half of our study region had FY-3C soil moisture observation data were used in our evaluation and analysis. To temporally agree with the FY-3C data, in situ soil moisture and rainfall data of these dates were extracted, and the NDVI data were interpolated to the dates.

### 3.2. Four Statistical Indicators

The MD represents the bias, which is the systematic difference between the satellite soil moisture retrievals and in situ soil moisture measurements. The MD can be obtained using the following equation:

$$MD = \frac{\sum_{i=1}^{N}\left(mv_i^{sat} - mv_i^{is}\right)}{N}. \tag{3}$$

The RMSE represents the absolute difference or accuracy of the soil moisture retrievals relative to in situ soil moisture measurements, which can be calculated as:

$$RMSE = \sqrt{\frac{\sum_{i=1}^{N}\left(mv_i^{sat} - mv_i^{is}\right)^2}{N}} \tag{4}$$

where $mv_i^{sta}$ represents the satellite soil moisture retrieval ($m^3/m^3$), $mv_i^{is}$ is the in situ soil moisture measurement ($m^3/m^3$), N represents the total number of samples, and i represents a specific sample. For temporal analysis, N varied for each grid cell and only dates that had valid data from both datasets were used for calculation. For spatial analysis, N varied for each date and only stations that had valid data from both datasets were used for calculation.

To better evaluate the estimation of the absolute difference of the satellite soil moisture products, we adopted the ubRMSE, which removes the bias of RMSE that characterizes random error. The ubRMSE is calculated using the following equation [48]:

$$ubRMSE = \sqrt{RMSE^2 - MD^2}. \tag{5}$$

The correlation coefficient R indicates the relative accuracy between the satellite soil moisture data and in situ soil moisture measurements. The R between the satellite soil moisture data and in situ soil moisture can be expressed as the following formula:

$$R = \frac{\sum_{i=1}^{N} \left(mv_i^{sat} - \overline{mv}^{sat}\right)\left(mv_i^{is} - \overline{mv}^{is}\right)}{(N-1)\sigma^{sta}\sigma^{is}} \tag{6}$$

where $\overline{mv}^{sat}$ is the satellite soil moisture average ($m^3/m^3$) during the whole evaluation period within one grid for temporal analysis, or of the valid stations in 1 day for spatial analysis; $\overline{mv}^{is}$ indicates the average of in situ soil moisture measurements ($m^3/m^3$); and $\sigma^{sat}$ and $\sigma^{is}$ are the standard deviation of satellite and in situ soil moisture ($m^3/m^3$), respectively.

## 4. Results

In this section, the statistical accuracy indicators of FY-3C, AMSR2, and SMAP SM retrievals are presented for both temporal and spatial scales. During the comparison, the in situ SM measurements from CASMOS were treated as the ground truth for all of the satellite SM products. The nighttime microwave satellite data were generally expected to produce more accurate soil moisture estimates than the daytime data. However, many previous studies proved that there were no significant differences between the soil moisture from daytime and nighttime overpasses [2,49]. Thus, we just ignored the daily discrepancies in this study. In Section 4.1, we examine the temporal performance of the FY-3C L2, AMSR2, and SMAP L3 products during the one-year period for each footprint in our study region; in Section 4.2, we evaluate the spatial performance of the footprints available for each date during the year.

### 4.1. Temporal Performance for Different Footprints

To understand the temporal agreement and consistency between the satellite soil moisture retrievals and in-situ measurements of different footprints, the four accuracy indicators for each valid footprint in our study area were computed separately. For example, we took the footprint that covered station O2342. As shown in Figure 5, the daily average soil moisture from FY-3C, AMSR2, and SMAP, as well as the monitoring station, were plotted in time over the entire assessment period. The three satellite datasets generally display different temporal variation patterns, during which the SMAP product shows the best accordance with that of ground observations. The four statistical indicators were calculated using the dates when the two compared datasets were both available. The error metrics of FY-3C, AMSR2, and SMAP for this footprint are summarized in Table 4. Note that the MD indicators were calculated by subtracting the in situ SM measurements from the satellite SM retrievals. A positive bias value indicates that the satellite soil moisture retrieval is larger (wetter) than the in-situ observation, whereas a negative value means that the satellite SM retrieval is lower (drier) than the in-situ observation.

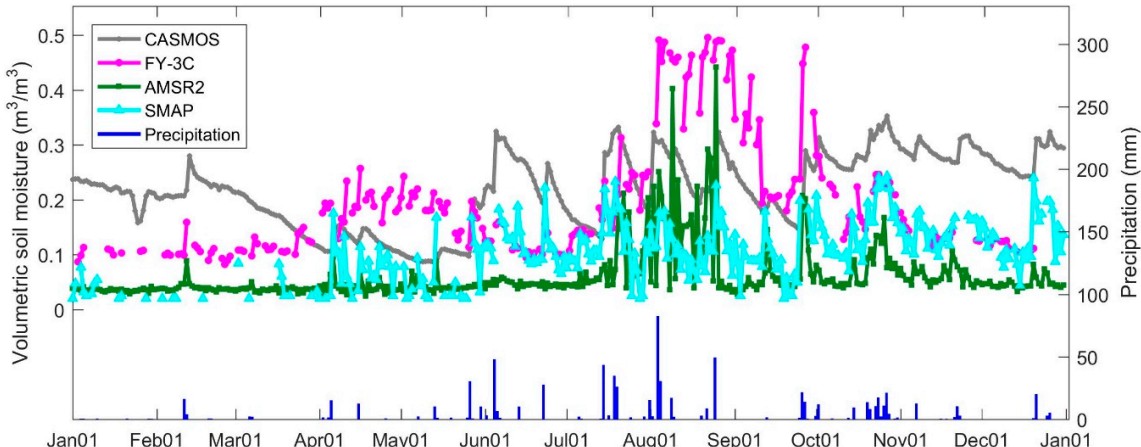

**Figure 5.** Temporal variations in the soil moisture data from the CASMOS stations and the FY-3C, AMSR2, and SMAP satellite products, and the rainfall data around the O2342 station during 2016. The gray line is the soil moisture (SM) daily mean from the CASMOS. The magenta, green, and cyan lines respectively represent the SM daily mean of the FY-3C, AMSR2, and SMAP products, and the blue histogram represents the daily precipitation.

**Table 4.** Statistical parameters for FY-3C, AMSR2, and SMAP against the CASMOS measurements of the O2342 station during 2016. Only the dates when the compared datasets both have observations were used.

| Products | MD ($m^3/m^3$) | RMSE ($m^3/m^3$) | ubRMSE ($m^3/m^3$) | R |
|----------|--------|----------|------------|------|
| AMSR2 | −0.15 | 0.17 | 0.07 | 0.27 |
| FY-3C | −0.02 | 0.12 | 0.11 | 0.21 |
| SMAP | −0.12 | 0.13 | 0.05 | 0.63 |

MD, mean difference; RMSE, root-mean-square error; ubRMSE, unbiased RMSE; R, correlation coefficient.

Using the abovementioned method, we then separately calculated the statistical indicators of the three satellite SM datasets against ground measurements for all the footprints. In total, 113 stations were used to estimate the statistical parameters. For each station, only the dates when the compared datasets both have observations were used. Figure 6 and Table 5 summarize the four statistical indicators for all the stations regardless of their location. Overall, the FY-3C soil moisture retrievals were drier than the in-situ measurements, with an average bias of −0.03 $m^3/m^3$. AMSR2 and SMAP show a drier bias than FY-3C, with an average value of −0.15 $m^3/m^3$ and −0.09 $m^3/m^3$, respectively. The dry bias may be relevant to the inconsistency of the sensing depths between the satellite and the ground measurements. The FY-3C L2 SM product shows a similar poor performance to the AMSR2 L3 product, with an average RMSE and ubRMSE value of 0.11 $m^3/m^3$ and 0.09 $m^3/m^3$, respectively, and an average R of 0.09. The SMAP L3 SM product demonstrates better performance, with an average RMSE and ubRMSE value of 0.12 $m^3/m^3$ and 0.06 $m^3/m^3$, respectively, and an average R of 0.49. SMAP can capture the temporal variations of near-surface soil moisture better than FY-3C and AMSR2. This result is consistent with our general expectation that the L-band microwave has a deeper sensing depth (approximately 5 cm) and is less susceptible to the influences of vegetation compared to higher frequencies, such as the C- and X-bands.

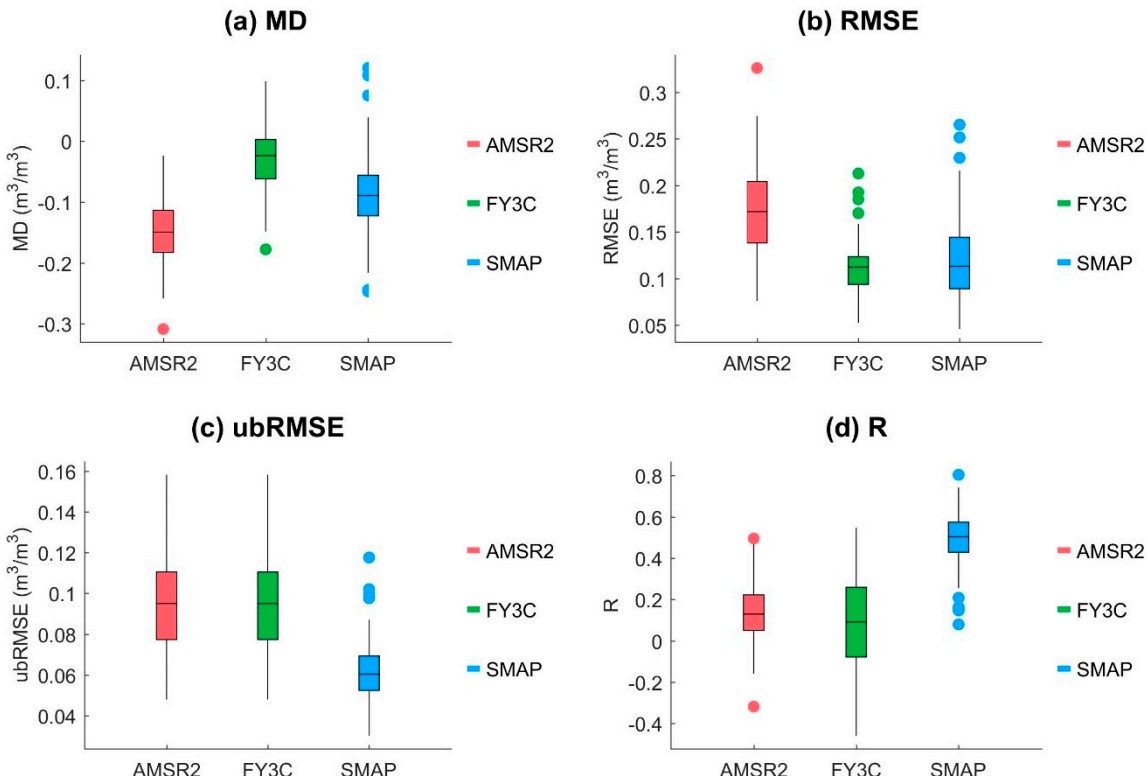

**Figure 6.** The temporal statistical indices between the satellite soil moisture datasets from FY-3C, AMSR2, and SMAP and the in-situ stations: (**a**) Mean difference (MD), (**b**) root-mean-square error (RMSE), (**c**) unbiased RMSE (ubRMSE), and (**d**) correlation coefficient (R). The median, the 1st quantile Q1, and the 3rd quantile Q3 are indicated by the box, the whiskers represent $Q1 - 1.5 (Q3 - Q1)$ and $Q3 + 1.5 (Q3 - Q1)$ values, and the points represent the outliers.

**Table 5.** The average statistical parameters for FY-3C, AMSR2, and SMAP against the CASMOS measurements of all of the footprints. One hundred and thirteen (113) stations were used for statistics.

| Products | MD ($m^3/m^3$) | RMSE ($m^3/m^3$) | ubRMSE ($m^3/m^3$) | R |
|----------|----------------|------------------|--------------------|---|
| AMSR2 | $-0.15$ | 0.17 | 0.09 | 0.14 |
| FY-3C | $-0.03$ | 0.11 | 0.09 | 0.09 |
| SMAP | $-0.09$ | 0.12 | 0.06 | 0.49 |

The above statistical parameters also indicate that the temporal consistency was different between different footprints even for the same satellite soil moisture product. To further illustrate how their consistency varied spatially, the statistical indicators of the FY-3C, AMSR2, and SMAP SM products were interpolated to the extent of our study area with a spatial resolution of 0.25° (Figure 7). The interpolations were carried out in ArcGIS software (Esri, NewYork, NY, USA) using the inverse distance weighted (IDW) method. In total, the footprints covering the 113 CASMOS stations (Figure 7) were used for the spatial interpolations.

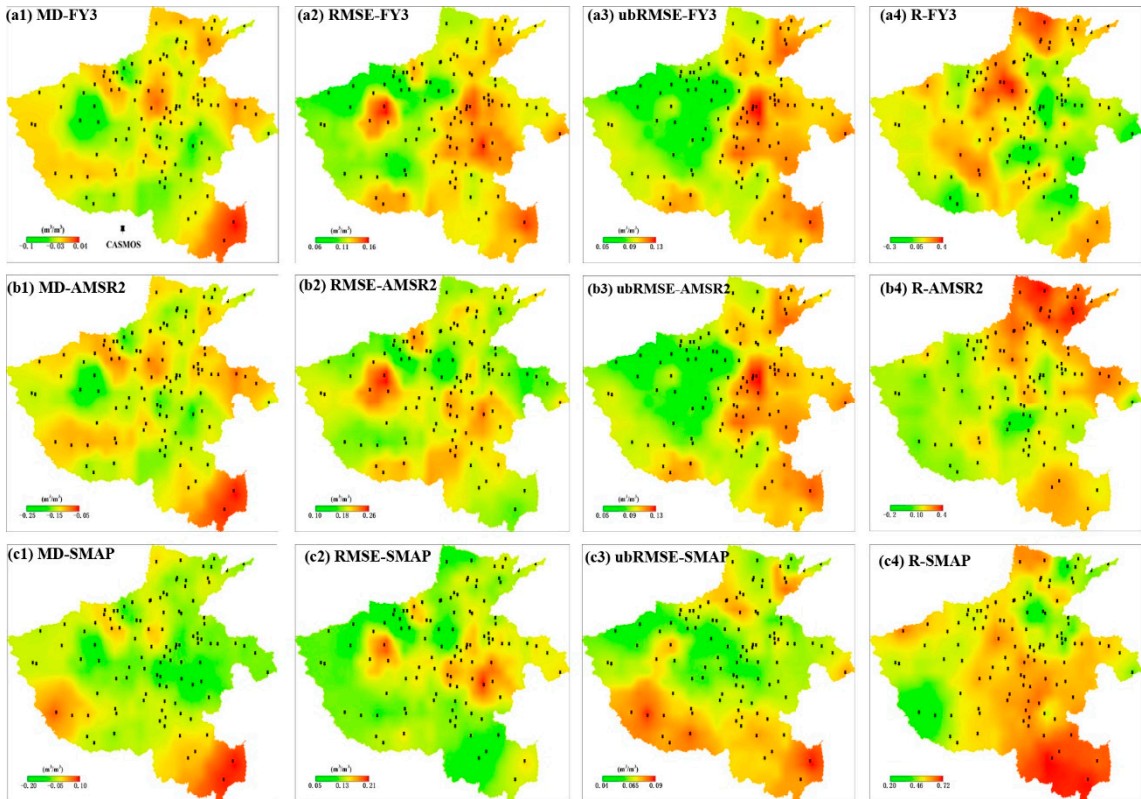

**Figure 7.** The spatial distribution difference between the FY-3C, AMSR2, and SMAP retrievals against in-situ measurements for the period of 1 January 2016 to 31 December 2016: (**a**) FY-3C, (**b**) AMSR2, and (**c**) SMAP. The black thumbtacks represent the 113 CASMOS stations used for the spatial interpolation. The interpolations were performed using the inverse distance weighted (IDW) method in ArcGIS, with the outsize as 0.25, the power as 1, and the search radius as 4 points.

As Figure 7 shows, for most grid cells of FY-3C, their biases were negative (see green and yellow colors in Figure 7a1), approximately ranging from –0.1 to –0.03 $m^3/m^3$. In terms of RMSE and ubRMSE, they shared a similar distribution pattern, in which most grid cell high values were located in the eastern part of Henan (Figure 7a2,a3), suggesting that the FY-3C SM retrievals were more consistent with in-situ measurements in the western part than in the eastern agricultural regions in Henan (Figure 1a). Similarly, the grid cells with higher correlation coefficients (red color in Figure 7a4) were located in western Henan. However, there were some exceptional regions. For example, in the southeast part of Henan a positive bias, larger RMSE and ubRMSE, as well as higher correlation were recorded. For the AMSR2 product, the MD, RMSE, and ubRMSE (Figure 7b1–b3) generally showed parallel distribution patterns like FY-3C, except that their values varied to an extent. The AMSR2 soil moisture product underestimated soil moisture for nearly all the grids, with a dry bias of −0.05 to −0.25 $m^3/m^3$, which was generally in accordance with the previous studies that the JAXA algorithm usually underestimates ground measurements [28,50,51]. The R of AMSR2 (Figure 7b4) indicated that it is more consistent with the in-situ measurements temporally in the northeast of Henan. The MD and RMSE of SMAP (Figure 7c1,c2) also showed similar patterns to those of FY-3C and AMSR2. We can see that the SMAP product outperformed the FY-3C product and the AMSR2 product in most of the study region, with a relatively smaller ubRMSE and higher correlation coefficients (Figure 7c3,c4).

*4.2. Spatial Performance At Different Times*

As shown in Figure 3, the NDVI and VWC variations in one year were dominated by the wheat–corn cropping system in our study region. In this section, we continued to evaluate how the

skill of the FY-3C, AMSR2, and SMAP soil moisture products vary with the NDVI and VWC variation at different times of the year. As shown in Figure 8, four dates (15 March 2016, 16 May 2016, 27 August 2016, and 27 October 2016) were first picked to display the typical performance of the satellite SM retrievals against in situ SM measurements (CASMOS) in different seasons. The figure indicated that the temporal variation was a key factor influencing the retrievals of the satellite SM products.

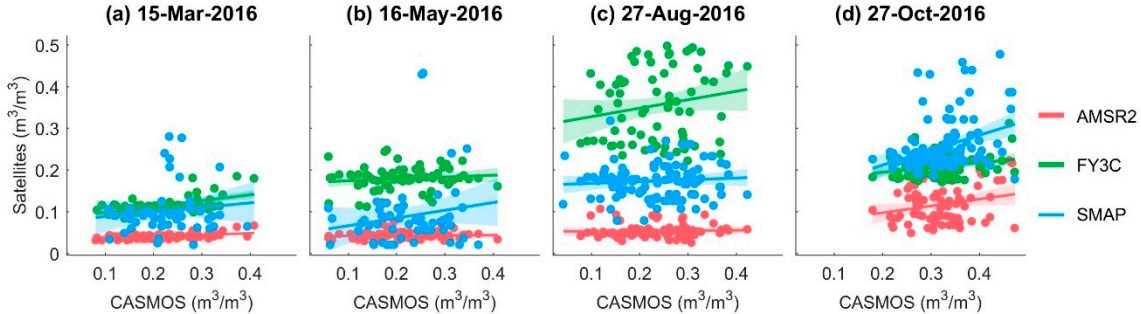

**Figure 8.** The soil moisture retrievals of the FY-3C, AMSR2, and SMAP products versus in situ SM measurements (CASMOS) on four different dates in 2016. The lines indicate the varying trend of the points.

We then calculated the statistical parameters of the three satellite products against in situ SM measurements of all the valid dates (if the satellite data cover half of the CASMOS stations within a day, we defined the day as a valid date) during the year. The number of valid days for FY3C, AMSR2, and SMAP was 233, 366, and 295, respectively. Figure 9 and Table 6 summarize the four indicators for all available dates, from which we can see that the consistency of the three satellite products was generally poor. For FY-3C, the average values of MD, RMSE, ubRMSE, and R were –0.06 $m^3/m^3$, 0.12 $m^3/m^3$, 0.07 $m^3/m^3$, and 0.22, respectively. However, compared with the temporal performance of the different footprints (Figure 6), the FY-3C SM L2 product showed better consistency with the CASMOS measurements on the spatial scale (Figure 9). For example, the average ubRMSE dropped from 0.09 to 0.07 $m^3/m^3$, and the average correlation coefficient R rose from 0.09 to 0.22. The AMSR2 product showed a much drier bias than FY-3C, which led to a large RMSE; however, like FY-3C, the spatial performance of AMSR2 was generally better than its temporal performance, with the average ubRMSE dropping from 0.09 to 0.07 $m^3/m^3$, and the average correlation coefficient R rising from 0.14 to 0.18. The SMAP product demonstrated a similar level of spatial performance as that of FY-3C and AMSR2, which is much worse than its temporal performance.

**Table 6.** The spatial statistical indices for FY-3C, AMSR2, and SMAP against the in-situ measurements.

| Products | MD ($m^3/m^3$) | RMSE ($m^3/m^3$) | ubRMSE ($m^3/m^3$) | R |
|----------|---------|-----------|-------------|------|
| AMSR2 | −0.16 | 0.18 | 0.07 | 0.18 |
| FY-3C | −0.06 | 0.12 | 0.07 | 0.22 |
| SMAP | −0.10 | 0.13 | 0.08 | 0.16 |

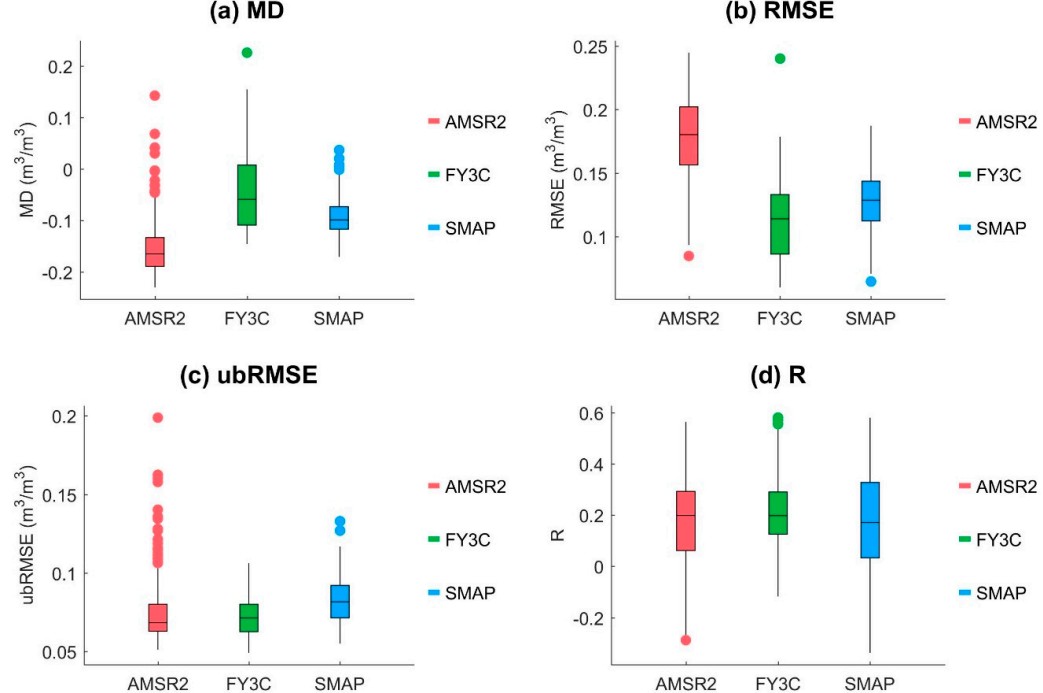

**Figure 9.** The spatial statistical indices of the FY-3C, AMSR2, and SMAP soil moisture datasets against the in-situ measurements for all of the valid dates during the year: (**a**) MD, (**b**) RMSE, (**c**) ubRMSE, and (**d**) R. The number of valid days of FY3C, AMSR2, and SMAP used for statistics was 233, 366, and 295, respectively.

Next, a more specific analysis was conducted to examine the temporal evolution of the statistical parameters between the compared datasets (Figure 10 and Table 7). As shown in the figure, the varying patterns of the four indicators were generally different. For FY-3C, the MD showed a double-peak trend with peaks around May and August, which is consistent with the wheat–corn cropping system in Henan. Except for the peaks, basically on all dates, FY-3C SM retrievals showed a negative bias compared to the in-situ measurements, which was further indicated by the monthly mean bias data in Table 7. The AMSR2 also had the smallest bias in May, August, and September, but the SMAP did not display any apparent trend in its bias. The high RMSE and ubRMSE and the low correlations between the FY-3C retrievals and in-situ measurements throughout the year indicated their inconsistency nearly all the time. However, we still can capture the influences of the cropping system on these statistical parameters. For example, around May, August, and September when the ground vegetation reached their maximum, FY-3C had a relatively large RMSE and ubRMSE, and a small R. We can also find that the AMSR2 statistical parameters shared a parallel trend with FY-3C in Table 7. The FY-3C and AMSR2 (JAXA algorithm) soil moisture products were all retrieved using the X-band brightness temperature [52], which may partly explain their similar performance. The SMAP statistical parameters did not show a similar seasonal variation like FY-3C and AMSR2.

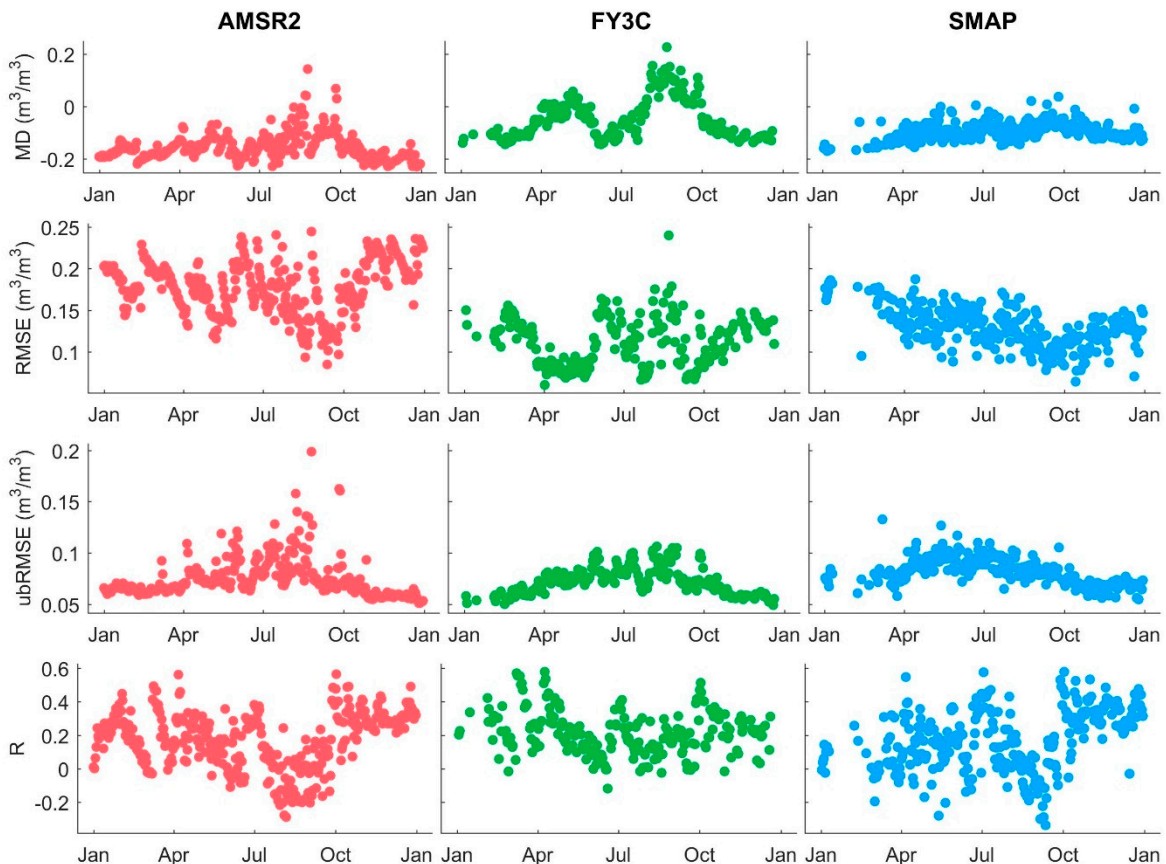

**Figure 10.** The temporal evolution of the statistical parameters of FY-3C, AMSR2, and SMAP in 2016.

**Table 7.** Monthly averaged statistics for FY-3C, AMSR2, and SMAP in 2016. Mean difference (MD), root-mean-square error (RMSE), and unbiased RMSE (ubRMSE) are in $m^3/m^3$.

| Indicators | Products | Jan. | Feb. | Mar. | Apr. | May | Jun. | Jul. | Aug. | Sep. | Oct. | Nov. | Dec. |
|---|---|---|---|---|---|---|---|---|---|---|---|---|---|
| | AMSR2 | −0.17 | −0.18 | −0.17 | −0.15 | −0.13 | −0.18 | −0.14 | −0.09 | −0.09 | −0.17 | −0.21 | −0.20 |
| MD | FY3C | −0.12 | −0.12 | −0.09 | −0.02 | −0.01 | −0.11 | −0.05 | 0.11 | 0.05 | −0.08 | −0.12 | −0.12 |
| | SMAP | −0.16 | −0.14 | −0.13 | −0.11 | −0.08 | −0.10 | −0.08 | −0.09 | −0.06 | −0.08 | −0.10 | −0.11 |
| | AMSR2 | 0.19 | 0.19 | 0.18 | 0.17 | 0.15 | 0.20 | 0.17 | 0.16 | 0.13 | 0.18 | 0.21 | 0.21 |
| RMSE | FY3C | 0.13 | 0.13 | 0.11 | 0.08 | 0.09 | 0.14 | 0.10 | 0.14 | 0.09 | 0.10 | 0.13 | 0.13 |
| | SMAP | 0.18 | 0.16 | 0.15 | 0.14 | 0.13 | 0.14 | 0.13 | 0.13 | 0.10 | 0.10 | 0.12 | 0.13 |
| | AMSR2 | 0.07 | 0.06 | 0.07 | 0.08 | 0.08 | 0.08 | 0.09 | 0.10 | 0.08 | 0.07 | 0.06 | 0.06 |
| ubRMSE | FY3C | 0.05 | 0.06 | 0.06 | 0.07 | 0.08 | 0.08 | 0.08 | 0.09 | 0.07 | 0.07 | 0.06 | 0.06 |
| | SMAP | 0.08 | 0.07 | 0.08 | 0.09 | 0.10 | 0.09 | 0.09 | 0.09 | 0.08 | 0.07 | 0.07 | 0.07 |
| | AMSR2 | 0.23 | 0.19 | 0.18 | 0.24 | 0.15 | 0.11 | 0.09 | -0.06 | 0.07 | 0.30 | 0.30 | 0.31 |
| R | FY3C | 0.26 | 0.24 | 0.30 | 0.34 | 0.17 | 0.11 | 0.20 | 0.14 | 0.17 | 0.30 | 0.21 | 0.18 |
| | SMAP | 0.07 | 0.10 | 0.06 | 0.17 | 0.14 | 0.14 | 0.23 | 0.03 | 0.01 | 0.32 | 0.35 | 0.31 |

## 5. Discussion

In this study, we investigated the estimated error of the remotely sensed SM products from FY-3C, along with AMSR2 and SMAP against in situ soil moisture measurements from the CASMOS on both temporal and spatial scales. The statistical indicators generally revealed that the FY-3C L2 SM product showed a poor consistency with the in situ SM data from CASMOS. The AMSR2 L3 SM product of JAXA algorithm exhibited a similar level of performance as FY-3C in our study region. The SMAP L3 SM product outperformed FY-3C and AMSR2 temporally, but showed lower performance in capturing the SM spatial variation. Apart from examining the accuracy of the three remotely sensed soil moisture products, we also investigated the potential factors that might influence the performance of the soil moisture products.

First, as conventionally performed in previous studies [2–4,16], during our analysis, the in situ soil moisture measurements from point stations were used as ground truth to evaluate the satellite soil moisture retrievals. However, with point-scale validation data, there may be several limitations during comparisons with more considerable footprint-scale satellite data. The monitoring stations supply soil moisture measurements at point locations, whereas the microwave sensors aboard satellites measure the average soil moisture within one satellite footprint. Due to the coarse resolution of the satellite products (25 km for FY-3C and AMSR2, 36 km for SMAP) and the spatial heterogeneity of the surface soil moisture, we can hardly use point-based in-situ measurements to correctly represent the spatially averaged soil moisture within a large satellite footprint [53]. Second, the sensing depth mismatch between in situ soil moisture and satellite observations may also contribute lots of uncertainties to the assessment results. Commonly, the effective soil moisture sensing depths at the L- and C/X-bands are 0–5 cm and 0–1 cm, respectively, which also depend on soil moisture [28,51]. The in situ soil moisture used in our evaluation was sourced from the station sensors deployed at a depth of 10 cm below the soil surface. FY-3C and AMSR2 utilized the X-band-observed brightness temperatures, while SMAP employed the L-band observations to retrieve their soil moisture products, which implies that the retrievals may not dependably represent the soil moisture in much deeper layers than the sensing depth. Third, other factors, such as the possible errors in ground measurements, vegetation coverage, precipitation, and climate characteristics, also influence the evaluation results, and their temporal and dynamic variation would lead to different levels of performance in the spatiotemporal analysis [2]. Considering the above, we could not expect the FY-3C retrievals, as well as the other two comparison soil moisture products from AMSR2 and SMAP, to exactly match the in-situ measurements from the CASMOS monitoring stations even under ideal conditions [3].

Additionally, the limited parameterizations of the microwave radiative transfer model and the inaccurate correction of the perturbing factors (e.g., surface temperature, vegetation, and surface roughness) in the soil moisture retrieval model are generally thought to be the leading cause of the inconsistency [54]. Among the factors, vegetation was one of the most significant influencing the soil moisture retrieval accuracy. In vegetated regions, the vegetation canopy attenuates signals from soil surfaces, with this effect increasing at higher frequencies, leading to a reduced sensitivity of the brightness temperatures to soil moisture. Accordingly, the accurate correction of the influence of vegetation is crucial for retrieving reliable soil moisture estimations. The effects of vegetation are commonly represented by the vegetation optical depth. In the current FY-3C soil moisture algorithm, the VWC is used as a proxy to calculate the vegetation optical depth (Table 3). To obtain the estimated VWC at a global scale, the VWC is empirically evaluated (refer to Table 3) by using a global, 10-year averaged AVHRR NDVI in the soil moisture retrieval algorithm, but their vegetation-related products are not currently released to the public. As a result, we cannot examine the temporal variations of the vegetation optical depth used in the FY-3C soil moisture algorithm, and further investigate how it impacts on the soil moisture retrievals.

In previous works, the effects of vegetation water content (VWC) on microwave radiative transfer and the soil moisture retrieval skill have been intensively investigated. Many studies have shown that C- and X-band observations can only be used over regions where vegetation is not too dense, and L-band radiometry is capable of retrieving soil moisture over relatively dense canopies (up to 3–5 kg/m$^2$) [55]. For instance, Calvet, et al. [56] indicated that a statistical soil moisture retrieval algorithm using C- and X-bands did not perform well with approximately 0–3 kg/m$^2$ VWC. Sawada, et al. [57] conducted in-situ observations of microwave brightness temperature, VWC, and soil moisture and revealed that there are few correlations between microwave signal and surface soil moisture when the VWC is larger than 0.3 kg/m$^2$.

As the soil moisture products of FY-3C and AMSR2 were derived from the X-band observations, the increase in uncertainty in these two soil moisture products with increasing vegetation was theoretically expected and was confirmed by our following analysis (Figures 11 and 12, and Table 8). Figure 11 indicated how VWC influences the soil moisture retrievals of the three satellites. The VWC

was estimated from the MODIS NDVI using the relationship (Equations (1) and (2)) proposed by Gao, et al. [46]. We used 0.3 kg/m$^2$ as a threshold value to divide the ground VWC conditions. As shown in the figure, for FY-3C and AMSR2, nearly all of the soil moisture with high values were retrieved under a high vegetation condition (VWC > 0.3 kg/m$^2$). However, for the soil moisture of SMAP, which was derived using the L-band observations, there were no apparent differences between the two different vegetation conditions. We further compared the temporal performance of the soil moisture products under the two different vegetation conditions. From Figure 12, we can see that under the low vegetation condition (VWC > 0.3 kg/m$^2$), the soil products generally showed better consistency with in-situ measurements with lower RMSE and ubRMSE and higher correlations. It is apparent to see the improvements for the FY-3C and AMSR2 products; e.g., when VWC < 0.3 kg/m$^2$, the ubRMSE of FY-3C dropped to 0.05 m$^3$/m$^3$ from 0.1 m$^3$/m$^3$, and the R rose to 0.29 from 0.14. It should be noted that the performance of the SMAP soil moisture product also exhibited some improvements under the lower vegetation condition, with the ubRMSE dropping from 0.06 m$^3$/m$^3$ to 0.05 m$^3$/m$^3$ and the R rising from 0.45 to 0.53. These results were also consistent with the consensus that the L-band is more sensitive to the surface soil moisture, especially in densely vegetated regions [2,37,56,57].

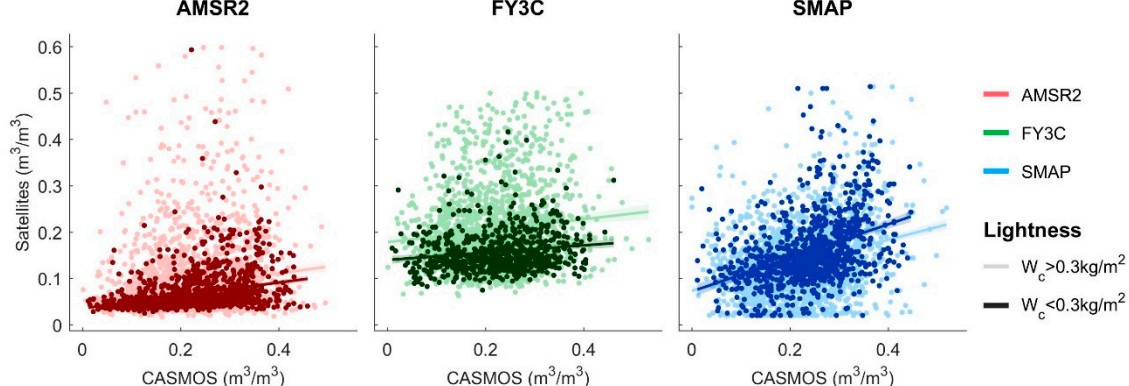

**Figure 11.** Comparison of the estimated soil moistures from FY-3C, AMSR2, and SMAP against in-situ measurements under different vegetation water content (VWC, W$_c$) conditions. VWC was determined using Equations (1) and (2).

**Table 8.** The statistical indicators for FY-3C, AMSR2, and SMAP soil moisture retrievals under the two contrasting vegetation water content (VWC, W$_c$) conditions.

| Products | VWC<0.3 kg/m$^2$ | | | | VWC>0.3 kg/m$^2$ | | | |
|---|---|---|---|---|---|---|---|---|
| | MD (m$^3$/m$^3$) | RMSE (m$^3$/m$^3$) | ubRMSE (m$^3$/m$^3$) | R | MD (m$^3$/m$^3$) | RMSE (m$^3$/m$^3$) | ubRMSE (m$^3$/m$^3$) | R |
| AMSR2 | −0.17 | 0.18 | 0.05 | 0.26 | −0.14 | 0.16 | 0.10 | 0.18 |
| FY-3C | −0.08 | 0.11 | 0.05 | 0.29 | −0.01 | 0.11 | 0.10 | 0.14 |
| SMAP | −0.09 | 0.11 | 0.05 | 0.53 | −0.09 | 0.12 | 0.06 | 0.45 |

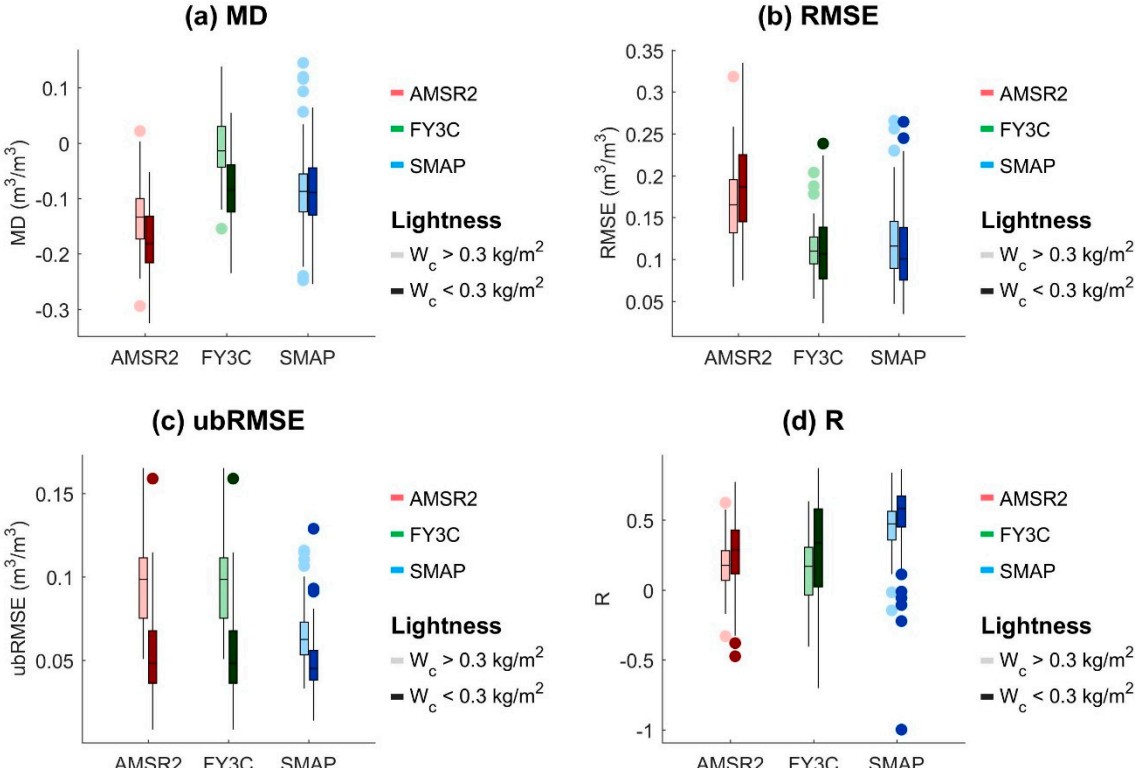

**Figure 12.** The temporal performance of the soil moisture products from FY-3C, AMSR2, and SMAP under the two different vegetation water content (VWC, W$_c$) conditions. VWC was estimated using Equations (1) and (2). In total, 113 stations were used to assess the statistical parameters. The VWC of about one-third of the valid dates was lower than 0.3 kg/m$^2$.

The assessment results (Figure 10 and Table 7) also revealed that the spatial performance at different times for the FY-3C soil moisture products exhibited a seasonal variation that was parallel to the cropping system in Henan. For example, the statistical parameters showed less dry bias, larger RMSE and ubRMSE, and smaller correlation coefficients around May, August, and September when the vegetation water content (VWC) of the crops reached their maximum. In Figure 13, we examined how the spatial performance of FY-3C, AMSR2, and SMAP varied with the average daily VWC. The VWC was empirically estimated from the MODIS NDVI data with Equations (1) and (2). From Figure 13a, we can see that the mean bias of FY-3C was highly related to the VWC. For AMSR2, the MD showed a positive but smaller correlation with the VWC. For SMAP, the MD was nearly unaffected by the VWC. These results further indicated that the L-band is less affected by the vegetation and more sensitive to the surface soil moisture [2,37,56,57]. Regarding RMSE and ubRMSE (Figure 13b,c), FY-3C also showed some positive correlation with VWC. Although the RMSE of AMSR2 decreased with the VWC, the ubRMSE showed a similar positive relationship with the VWC like FY-3C. As expected, the RMSE and ubRMSE of SMAP showed the smallest correlation with the VWC. The correlation coefficients (R) of FY-3C, AMSR2, and SMAP generally showed similar decreasing trends with increasing VWC. The above analyses together indicated that the ground vegetation water content has a considerable influence on the performance of the remotely sensed soil moisture products, especially for the X-band. That is to say, the impact of vegetation could be captured by using a time-series-based approach to soil moisture assessment. To improve the performance of the FY-3C soil moisture product, an improved algorithm that could better characterize the ground vegetation effects should be applied in the future.

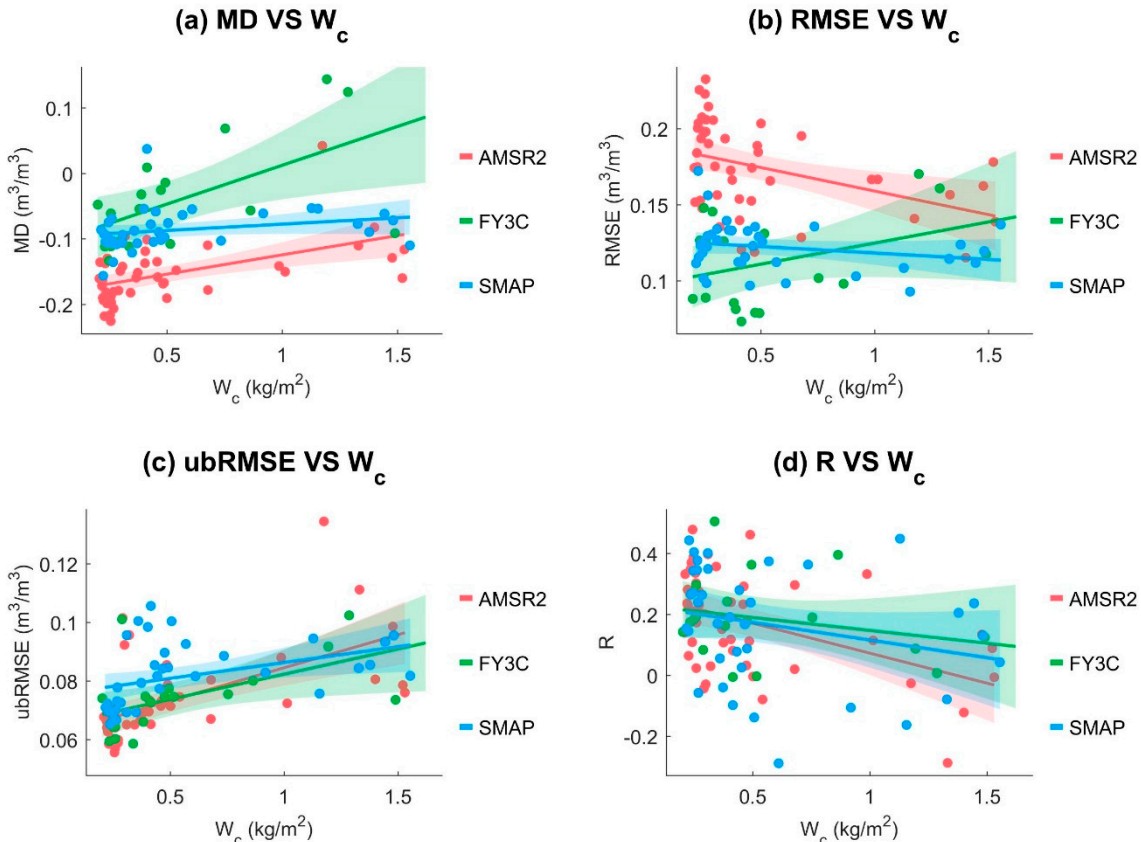

**Figure 13.** The spatial evaluation results (MD, RMSE, ubRMSE, and R) versus VWC at different times in the year. VWC was estimated from the MODIS NDVI data using the empirical relationship; refer to Equations (1) and (2). Only the 46 dates when there existed NDVI data were used.

## 6. Conclusions

We evaluated the FY-3C Level 2 daily soil moisture product in Henan province in China using in-situ data from 113 soil moisture monitoring stations deployed by the CMA Meteorological Observation Centre and contrasted it with the retrieval skill of the soil moisture products of AMSR2 and SMAP. The assessment results revealed that the FY-3C L2 SM product showed a poor consistency with the in situ SM data from CASMOS. If the in-situ measurements were treated as ground truths, the absolute accuracy of FY-3C soil moisture retrievals was 0.12 $m^3/m^3$ (RMSE), which is much worse than the desired accuracy of 0.06 $m^3/m^3$. Also, the AMSR2 L3 SM product of the JAXA algorithm exhibited a similar level of performance as FY-3C in our study region. The SMAP L3 SM product outperformed FY-3C and AMSR2 temporally, but showed lower performance in capturing the SM spatial variation.

The FY-3C L2 SM product tended to overestimate the soil moisture amount when the crop biomass is large in May, August, and September and underestimates soil moisture during the rest of the year. This result agrees with our expectation because vegetation water considerably influences passive microwave soil moisture retrievals in the footprint. In conclusion, the accuracy and reliability of the FY-3C soil moisture estimates in agricultural areas depend upon the crop types as well as their growing stages. This issue should be addressed in future studies to improve the accuracy of FY-3C soil moisture estimates.

**Author Contributions:** Conceptualization, Y.Z. and S.F.; Formal analysis, Y.Z., X.L., D.W., and R.S.; Funding acquisition, S.P. and S.F.; Methodology, Y.Z., D.W., and R.S.; Project administration, S.P. and S.F.; Resources, S.P. and S.F.; Writing (original draft), Y.Z.; Writing (review & editing), Y.Z., S.P., S.J., J.W., X.L., D.W., R.S., and S.F.

**Funding:** This research was funded by the Project2 of International Cooperation and Exchanges NSFC (NSFC-RCUK_STFC) (61661136005), the NSFC Program (41490633), and the UK STFC Program (ST/N006836/1).

**Acknowledgments:** The authors wish to thank NSMC, CMDSC, and NSDIC for making the soil moisture, precipitation, and NDVI data available online.

**Conflicts of Interest:** The authors declare no conflict of interest.

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
