# Peer review of "Evaluation of Fengyun-3C Soil Moisture Products Using In-Situ Data from the Chinese Automatic Soil Moisture Observation Stations: A Case Study in Henan Province, China"

_water, doi:10.3390/w11020248_

Round 1

Reviewer 1 Report

The article is quite interesting, English language and style are fine, but it is not very well written. There are some negative points: first of all, any information about the soils occurring in the study area is completely missing. How many point data were actually used for statistical elaborations? In my opinion a single year of observations is insufficient to fully evaluate the performance of satellite products, because there could be some anomalies in the considered year. In this paper you evaluated such performance only for 2016, and your results cannot be generalised. My greater concerns are about the interpolation in space of the four statistical parameters: which values were interpolated? An average along the whole year or a single measurement? This is not explained. But above all, applying the Kriging algorithm in ArcGIS as a blackbox, without a critical analysis of the dataset is a big conceptual error. To be correctly applied, Kriging requires that the data are normally distributed in space and spatially autocorrelated: neither statistical nor variographic analyses were conducted to verify these prerequisites in the dataset. Which parameters did you use in Kriging algorithm for the interpolation? Such parameters must be chosen from the experimental variogram. An alternative to this method, if it is not important the evaluation of the interpolation error in each point, is to use a deterministic interpolator like IDW.

 Other specific comments:

1. All the acronyms must be defined when first appear in the text, even if of relatively common use.

2. Line 187: this is very unlikely: usually almost all the pedogenetic factors have the same importance in determining the surface soil. Maybe the word "moisture" is missing in this sentence?

3. Some papers are not correctly reported in the reference list, and for some others insufficient information to find them is provided.

Author Response

Response to Reviewer 1 Comments

Point 1: The article is quite interesting, English language and style are fine, but it is not very well written.

Response 1: Thanks for the reviewer’s positive comment about the English language and style of our manuscript. As a non-native English speaker, it’s quite challenging for us to finish it very well in one hit. To finish it better, we have had this manuscript gone through English editing provided by MDPI. At the same time, our co-authors from UK also helped to review it to try to improve the language problems.

Point 2: There are some negative points: first of all, any information about the soils occurring in the study area is completely missing.

Response 2: According to the reviewer’s suggestion, we have added some background information of the soil types in our study area. Please refer to section 2.1, lines: 142-155.

Point 3: How many point data were actually used for statistical elaborations?

Response 3: Sorry we didn’t give this information clearly in the last version manuscript. When evaluating the temporal performance of the soil moisture products of FY3C, AMSR2, and SMAP, 113 out of the 158 CASMOS stations were used to calculate the statistical parameters. The 45 observation stations located in cities were excluded. We noted this clearly in captions of the figures and tables this time. When evaluating their spatial performance, only the dates when FY3C, AMSR2, or SMAP covered at least 50% of the study region were used to get the statistical parameters of their corresponding soil product. The valid days for FY3C, AMSR2, and SMAP were 233, 366, and 295, respectively. Please refer to lines: 371-372, and 438-439 and Table 5 and Figure 7, and Figure 9.

Point 4: In my opinion a single year of observations is insufficient to fully evaluate the performance of satellite products, because there could be some anomalies in the considered year. In this paper you evaluated such performance only for 2016, and your results cannot be generalised.

Response 4: We agreed with the reviewer that a multi-year of dataset will give us more confidence to filter out the anomalies where the bias is not relevant, and further to evaluate the performance of the soil moisture products. It is indeed the place where our work should strengthen in the future. However, due to the CASMOS data sharing limitation, until now we just got the in situ data of 2016. To reduce the uncertainty in our evaluation results of FY3C SM product, we also compared the SM products from AMSR2 and SMAP with the in situ soil moisture from the CASMOS stations. 

Point 5: My greater concerns are about the interpolation in space of the four statistical parameters: which values were interpolated? An average along the whole year or a single measurement? This is not explained.

Response 5: Sorry we didn’t clearly explain which values we used for interpolations in the figures. The footprints which cover the 113 CASMOS stations were used for the interpolation in space of the four statistical parameters. The statistical parameters of each footprint were calculated using all the measurements available in the year. We have added a note to explain it accordingly in the manuscript, please refer to lines: 394-397.

Point 6: But above all, applying the Kriging algorithm in ArcGIS as a blackbox, without a critical analysis of the dataset is a big conceptual error. To be correctly applied, Kriging requires that the data are normally distributed in space and spatially autocorrelated: neither statistical nor variographic analyses were conducted to verify these prerequisites in the dataset. Which parameters did you use in Kriging algorithm for the interpolation? Such parameters must be chosen from the experimental variogram. An alternative to this method, if it is not important the evaluation of the interpolation error in each point, is to use a deterministic interpolator like IDW.

Response 6: Many thanks for the reviewer’s great suggestions. We didn’t notice the strict conditions when applying the Kriging interpolation algorithm. Considering the reviewer’s suggestions, we replotted all the interpolation figures with the IDW method in ArcGIS. Please refer to Figure 7 in the manuscript.

Point 7: All the acronyms must be defined when first appear in the text, even if of relatively common use.

Response 7: According to the reviewer’s suggestion, we have double checked the manuscript to ensure all the acronyms have been defined when they first appear in the text.

Point 8: Line 187: this is very unlikely: usually almost all the pedogenetic factors have the same importance in determining the surface soil. Maybe the word "moisture" is missing in this sentence?

Response 8: As the reviewer suggested, the sentence is more reasonable with the word “moisture”. We have revised it accordingly in our manuscript. Please refer to line 267.

Point 9: Some papers are not correctly reported in the reference list, and for some others insufficient information to find them is provided.

Response 9: Many thanks that the reviewer noticed the problems of our references. We rechecked all the papers we referred to in our manuscript. It’s true that some of them can’t be retrieved in google scholar. We found that these papers were published in some Chinese journals in Chinese and their titles and abstracts were translated into English by the journals. So, aside from the necessary information water journal requests, we also added the URL where you can find the papers. Please refer to the reference list.

Reviewer 2 Report

In the current study, authors compare the soil moisture estimated from one Satellite and found that the Satellite is unable to capture the soil moisture except for few months. The authors should utilize the soil moisture data available from other satellites e.g. SMOS and SMAP, AMSR-2 and compare with the field estimated data and find out which satellite/sensor is able to capture the soil moisture.

Author Response

Response to Reviewer 2 Comments

Point 1: In the current study, authors compare the soil moisture estimated from one Satellite and found that the Satellite is unable to capture the soil moisture except for few months. The authors should utilize the soil moisture data available from other satellites e.g. SMOS and SMAP, AMSR-2 and compare with the field estimated data and find out which satellite/sensor is able to capture the soil moisture.

Response 1: Many thanks for the reviewer’s constructive comments. Considering the reviewer’s suggestions, we analysed the soil moisture data from SMAP and AMSR-2 against the in-situ soil moisture measurements, and we also compared their evaluation results with FY3C soil moisture product in our revised manuscript. Due to the Radio Frequency Interference (RFI) problem of SMOS in China, we didn’t consider its soil moisture product during our comparison.

In the manuscript, we rewrote some parts of the content and replotted most of the figures accordingly. Please refer to the revised manuscript. Thanks again!

Reviewer 3 Report

In this paper, the FY-3C soil moisture retrieval product was evaluated by in-situ soil moisture data from the Chinese automatic observation stations. The authors found the low skill of the FY-3C product to reproduce in-situ surface soil moisture observations. Vegetation effects may degrade the skill of the FY-3C soil moisture product.

General comments:

The topic of this paper is suitable to water and this paper is generally well written. However, I believe that there are many things to do in order to reach the full potential of this paper. I recommend the editor to reconsider this paper after major revision.

First, the new scientific finding of this paper is unclear. It is extremely important to validate the accuracy of the new satellite observation using in-situ observations. However, this paper has few scientific contributions to the published literature and it seems to be a technical report rather than a scientific paper. What is the take home message of this paper, which is even useful for researchers outside the FY-3C community?

Second, I believe that the authors may compare the skill of the FY-3C soil moisture product with that of the other satellite datasets such as AMSR2, SMOS, and SMAP. The authors indicated that the FY-3C soil moisture product is poor at reproducing the in-situ observations. If the skill of the other satellite products is not so poor, there might be some problems in the FY-3C microwave observation (i.e. no L- and C-bands observation) and/or the retrieval algorithm. Please clarify how poor the FY-3C product is compared with the other satellite products.

Third, vegetation effects should be investigated more deeply and quantitatively. In previous works, the effects of vegetation water content (VWC) on microwave radiative transfer and the soil moisture retrieval skill have been intensively investigated. For instance, Calvet et al. (2011) indicated that a statistical soil moisture retrieval algorithm using C- and X-bands did not perform well with approximately 0-3 [kg/m2] VWC. Sawada et al. (2017) performed the in-situ observations of microwave brightness temperature, VWC, and soil moisture and revealed that there are few correlations between microwave signal and surface soil moisture when VWC is larger than 0.3 [kg/m2]. It is generally difficult to directly observe VWC but Gao et al. (2015) provided the robust relationship between NDVI and VWC so that the authors estimate VWC in their study area. Please refer these previous works and discuss if the authors’ findings are correlated with them. In addition, please show the skill of soil moisture retrieval as a function of NDVI. Is the skill of soil moisture retrieval accurate in the pixels with low NDVI? I believe that this point is indeed unclear and Figure 10 does not help.

The other point-by-point comments are shown below.

Major Points:

L130: Please note that there are several AMSR-E soil moisture products and retrieval algorithms. I believe that the authors’ algorithm is based on LPRM. Please explicitly describe that their algorithm is similar to LPRM. In addition, how similar is their algorithm to LPRM? Please explicitly describe the differences between LPRM and the algorithm used in this paper.

Minor Points:

L182: What did the authors tell by Figure 2? I believe that Figure 2 is unnecessary.

References

Calvet, J.C.; Wigneron, J.P.; Walker, J.; Karbou, F.; Chanzy, A.; Albergel, C. Sensitivity of Passive Microwave Observations to Soil Moisture and Vegetation Water Content: L-Band to W-Band. IEEE Trans. Geosci. Remote Sens. 2011, 49, 1190–1199.

Sawada, Y.; Tsutsui, H.; Koike, T. Ground Truth of Passive Microwave Radiative Transfer on Vegetated Land Surfaces. Remote Sens. 2017, 9, 655.

Author Response

Response to Reviewer 3 Comments

Point 1: First, the new scientific finding of this paper is unclear. It is extremely important to validate the accuracy of the new satellite observation using in-situ observations. However, this paper has few scientific contributions to the published literature and it seems to be a technical report rather than a scientific paper. What is the take home message of this paper, which is even useful for researchers outside the FY-3C community?

Response 1: Many thanks for the reviewer’s constructive comments. The reviewer raised a very good and challenging question. To answer this question, first we compared the skill of the soil moisture products from SMAP (L3 36 km) and AMSR2 (JAXA) with FY-3C. From the statistical parameters, we found that AMSR2 and FY-3C generally showed a similar skill level; Whereas, the skill of SMAP soil moisture product is much better than AMSR2 and FY3C. Our results are consistent with the consensus that L-band observations (SMAP) are generally thought to be more sensitive to soil moisture than higher frequencies, like C- and X-bands (AMSR2, FY3C). Also, considering the reviewer’s suggestions, we estimated the ground VWC using the MODIS NDVI and we analysed the performance of the soil moisture products of the three satellites under different VWC conditions. Our results showed that under lower VWC condition, the soil moisture products exhibited much better performance than high VWC condition and their performance decreased with the increased VWC. We believed that these complementary works will further enhance the scientific finding of this paper. Thanks again for your constructive suggestions.

Point 2: Second, I believe that the authors may compare the skill of the FY-3C soil moisture product with that of the other satellite datasets such as AMSR2, SMOS, and SMAP. The authors indicated that the FY-3C soil moisture product is poor at reproducing the in-situ observations. If the skill of the other satellite products is not so poor, there might be some problems in the FY-3C microwave observation (i.e. no L- and C-bands observation) and/or the retrieval algorithm. Please clarify how poor the FY-3C product is compared with the other satellite products.

Response 2: Considering the reviewer’s suggestions, we analysed the soil moisture data from SMAP (L3 36km) and AMSR-2 (JAXA) against the in-situ soil moisture measurements, and we also compared their evaluation results with FY-3C soil moisture product in our revised manuscript. Due to the man-made Radio Frequency Interference (RFI) contamination problem of SMOS in China, we didn’t consider its soil moisture product during our comparison. In the revised manuscript, we rewrote most of the contents and replotted the figures accordingly. Please refer to the revised manuscript.

Point 3: Third, vegetation effects should be investigated more deeply and quantitatively. In previous works, the effects of vegetation water content (VWC) on microwave radiative transfer and the soil moisture retrieval skill have been intensively investigated. For instance, Calvet et al. (2011) indicated that a statistical soil moisture retrieval algorithm using C- and X-bands did not perform well with approximately 0-3 [kg/m2] VWC. Sawada et al. (2017) performed the in-situ observations of microwave brightness temperature, VWC, and soil moisture and revealed that there are few correlations between microwave signal and surface soil moisture when VWC is larger than 0.3 [kg/m2]. It is generally difficult to directly observe VWC but Gao et al. (2015) provided the robust relationship between NDVI and VWC so that the authors estimate VWC in their study area. Please refer these previous works and discuss if the authors’ findings are correlated with them. In addition, please show the skill of soil moisture retrieval as a function of NDVI. Is the skill of soil moisture retrieval accurate in the pixels with low NDVI? I believe that this point is indeed unclear and Figure 10 does not help.

Response 3: According to the reviewer’s suggestions, we first estimated the VWC using the empirical equations proposed by Gao et al. (2015) (lines: 255-266 and Figure 3). Then, we analysed how the soil moisture products performed under two different VWC conditions (i.e., >0.3 kg/m2 and <0.3 kg/m2). The soil moisture products showed much better skill under low VWC conditions. We further analysed the variation trends of the statistical parameters versus the VWC at different times of the year, which indicated that the performance of the soil products decreased with the increased VWC. These results were generally consistent with the findings of the previous studies. Please refer to lines: 525-589 in Discussion of the revised manuscript.

Point 4: L130: Please note that there are several AMSR-E soil moisture products and retrieval algorithms. I believe that the authors’ algorithm is based on LPRM. Please explicitly describe that their algorithm is similar to LPRM. In addition, how similar is their algorithm to LPRM? Please explicitly describe the differences between LPRM and the algorithm used in this paper.

Response 4: Sorry we didn’t explicitly describe the algorithm of the FY3C soil products in last manuscript. This time, we consulted the FY3C soil moisture product team and we found the algorithm used in FY3C was more similar to the Single Channel Algorithm (SCA) model proposed by Jackson (1993). In section 2.2 of the revised manuscript, we give a detailed description of the parameterization of the FY3C algorithm. Refer to lines 84 to 90, and 172 to 194.

Point 5: L182: What did the authors tell by Figure 2? I believe that Figure 2 is unnecessary.

Response 5: We updated the Figure with a new one. The new figure showed the averaged NDVI and VWC of all the CASMOS stations in 2016. Please refer to Figure 3, lines 262-266.

Round 2

Reviewer 3 Report

I think that the authors addressed all of my comments and signifincantly improved the paper.

I have a small editing comment. In the caption of Figure 7, the authors wrote "(a) MD, (b) RMSE, ....." but I believe it should be "(a) FY3, (b) AMSR2, .....

Basically, I believe that this paper is ready for publication.

Author Response

Point 1: I have a small editing comment. In the caption of Figure 7, the authors wrote "(a) MD, (b) RMSE, ....." but I believe it should be "(a) FY3, (b) AMSR2, .....

Response 1: Sorry for this careless mistake. We have revised it accordingly in the revised manuscript (lines: 423-424). Thanks again for your great help for improving this paper.
